# Target Selection for T-Cell Therapy in Epithelial Ovarian Cancer: Systematic Prioritization of Self-Antigens

**DOI:** 10.3390/ijms24032292

**Published:** 2023-01-24

**Authors:** Paul Schossig, Ebru Coskun, Ruza Arsenic, David Horst, Jalid Sehouli, Eva Bergmann, Nadine Andresen, Christian Sigler, Antonia Busse, Ulrich Keller, Sebastian Ochsenreither

**Affiliations:** 1Department of Hematology, Oncology and Cancer Immunology, Campus Benjamin Franklin, Charité-Universitätsmedizin Berlin, 10117 Berlin, Germany; 2German Cancer Consortium (DKTK), German Cancer Research Center (DKFZ), 69120 Heidelberg, Germany; 3Department of Pathology, Universitätsklinikum Heidelberg, Heidelberg University, 69120 Heidelberg, Germany; 4Insitute of Pathology, Charité-Universitätsmedizin Berlin, 10117 Berlin, Germany; 5Department of Gynecology, Charité-Universitätsmedizin Berlin, 10117 Berlin, Germany; 6Tumorbank Ovarian Cancer Network, 13353 Berlin, Germany; 7Charité Comprehensive Cancer Center, Charité-Universitätsmedizin Berlin, 10117 Berlin, Germany; 8Max-Delbrück-Center for Molecular Medicine, 13125 Berlin, Germany

**Keywords:** immunotherapy, tumor associated antigen, ovarian cancer, cytotoxic T lymphocytes, KIF20A, CT45, LY6K, Cyclin A1

## Abstract

Adoptive T cell-receptor therapy (ACT) could represent a promising approach in the targeted treatment of epithelial ovarian cancer (EOC). However, the identification of suitable tumor-associated antigens (TAAs) as targets is challenging. We identified and prioritized TAAs for ACT and other immunotherapeutic interventions in EOC. A comprehensive list of pre-described TAAs was created and candidates were prioritized, using predefined weighted criteria. Highly ranked TAAs were immunohistochemically stained in a tissue microarray of 58 EOC samples to identify associations of TAA expression with grade, stage, response to platinum, and prognosis. Preselection based on expression data resulted in 38 TAAs, which were prioritized. Along with already published Cyclin A1, the TAAs KIF20A, CT45, and LY6K emerged as most promising targets, with high expression in EOC samples and several identified peptides in ligandome analysis. Expression of these TAAs showed prognostic relevance independent of molecular subtypes. By using a systematic vetting algorithm, we identified KIF20A, CT45, and LY6K to be promising candidates for immunotherapy in EOC. Results are supported by IHC and HLA-ligandome data. The described method might be helpful for the prioritization of TAAs in other tumor entities.

## 1. Introduction

According to estimates by the International agency for research of cancer, ovarian cancer appears to be the most lethal malignancy of the female reproductive system with roughly 295,000 new cases causing over 185,000 cases of death in 2018 worldwide [1,2] with epithelial ovarian carcinomas (EOC) being responsible for the majority of cases (~90%) [3]. Due to the lack of symptoms in early stages, its common peritoneal spread, and insufficient measures of early detection, about 60% of patients are diagnosed with EOC in a metastatic stage [4]. Although the outcome of first-line therapy, consisting of cytoreductive surgery followed by chemotherapy with platinum and taxane, has improved by the addition of PARP and angiogenesis inhibitors [5,6,7], the overall survival (OS) remains dismal with a 5-year survival rate between 40–49% [4]. Modern unspecific immune therapies in form of modulation of checkpoint molecules showed limited activity and did not improve OS. In this scenario, one promising alternative is the field of targeted T cell-based immunotherapy.

The development and clinical application of targeted T cell therapies of cancer, which can be applied in form of adoptive cell therapy (ACT) or vaccination strategies, often combined with an unspecific immunotherapy, have made immense progress. The majority of ACT approaches are conducted as adoptive transfer of autologous T cells with a transgenic, tumor associated antigen (TAA)-specific T cell receptor (TCR), a strategy, which is considered the most effective option to apply a targeted T cell therapy in cancer [8,9]. For this purpose, autologous T cells are collected, in vitro transfected with a TCR that specifically recognize an epitope against a predefined TAA in context with a common HLA molecule, expanded in vitro, and transferred back to the patient to elicit a pseudomonoclonal T cell response [10,11,12]. Further targeted T cell strategies include vaccination and TCR-based bispecific T cell engagers [13,14]. Compared to strategies targeting structures on the cell surface such as chimeric antigen receptor (CAR) T cells or antibody-based approaches, the use of TCR-based strategies poses the advantage that it can target intracellular TAA. A clear disadvantage is the limitation in patient elegibility to such a treatment by the HLA restriction [15]. The clinical effectiveness of ADT is not determined by inflammation or spontaneous T cell infiltrates, but mostly by the selective expression of the respective TAA in the tumor.

EOC is a tumor entity potentially highly amenable for a targeted T cell therapy because (1) a multitude of TAA, especially cancer testis antigens (CTA) are expressed in high frequencies by the tumor, and (2) the course of the disease with several lines of palliative chemotherapy and subsequent episodes of follow-up offers the opportunity to either introduce an ACT approach as an additional treatment line to the therapeutic algorithm or to add ACT as a consolidation after a successful line of cytotoxic therapy. In recent years, many ACT trials targeting different TAAs such as NY-ESO1 or mesothelin in EOC have provided promising results, such as partial remission or prolonged disease stabilization [16,17,18,19,20,21]. However, some TAAs are only expressed in a small fraction of EOCs (e.g., ERBB2, GAGE) [22,23], while others (e.g., NY-ESO1) show intrapatient heterogeneous or unstable expression patterns [24,25] or are expressed in healthy tissue (e.g., WT1, MMP7), which can limit the therapeutic applicability or even hold the risk of adverse events in form of on target/off tumor toxicity [26].

The identification and selection of the TAA is a crucial step in the development of any targeted T cell therapy strategy. Features of a suitable TAA include the high and selective expression in the tumor in a high percentage of patients, the stable expression in a high percentage of tumor cells, functional relevance for the maintenance of the malignant phenotype, expression in the stem cell compartment as far as such a tumor-initiating cell population is defined for the respective tumor type, T cell immunogenicity, and a sufficient number of T cell epitopes. To facilitate this complex selection process, Cheever et al. created a list of criteria for the prioritization of available TAA even though no differentiation between TCR and antibody-based antigens was made [27]. These criteria were broken down into subcriteria and weighted, according to their relative importance as defined by a board of experts in the field of immunotherapy of cancer. This vetting algorithm not only resulted in a prioritization of all relevant TAA at that point in time, it also allowed to evaluate and contextualize new TAA candidates applying the weighted criteria.

The goal of this study is to systematically identify and prioritize TAAs for targeted T cell therapy of EOC. To achieve this, a comprehensive list of previously described TAAs is created by compiling the self-antigens of two cancer antigen databases. Neoantigens or virus-associated antigens were excluded, the first because in EOC neoantigens are highly patient specific and not widely clinically applicable, the latter because EOC has no association to oncogenic viruses. After exclusion of TAAs with high expression in healthy adult tissues and/or low expression in a TCGA-dataset of EOCs, we applied a modified version of the Cheever prioritization criteria on the remining candidates and applied additional data including IHC, clinical data, and published ligandome data. With this approach we identified top targets for T cell-based therapy in EOC and provide guidance for reevaluation as soon as sufficient clinical data will become available.

## 2. Results

### 2.1. Identification of Potential TAAs for EOC Therapy

The identification process of potential TAAs for EOCs constituted of three steps, as illustrated in Figure 1. The first step was the creation of a comprehensive list of previously described TAAs. A total of 237 previously described targets for immunotherapeutic approaches were identified, out of which 70 neoantigens were excluded. In order to evaluate the expression of TAA in healthy tissues, the remaining 167 candidates were then entered in the GTEx-portal and sorted into three groups dependent on median expression, ranging from low over medium to high expression. A total of 60 TAAs were sorted into the high group, deemed unsuitable for immunotherapy and were excluded. The 107 TAAs in the low and medium group, n = 60 and n = 47, respectively, were investigated for their expression in a TCGA-dataset of EOCs (n = 373). Median and average fragments per kilobase of transcript per million (FPKM) values of RNA-expression were used for evaluation. This selection step led to the exclusion of n = 69 candidates, which have either shown a lower expression in EOC samples than their respective cut-off value (n = 66) or there was no data on the expression in EOC available (n = 3). Out of the remaining 38 TAAs, n = 21 belonged to the low group, while n = 17 belonged to the medium group. These TAAs formed the set of the most promising candidates with negative or low expression in healthy tissues and relevant expression in EOC samples and were selected for the following evaluation process (Appendix A).

### 2.2. Evaluation and Prioritization of TAAs

To paint a more comprehensive picture of the 38 TAAs identified, MEDLINE database was searched via PubMed in May 2020, with the goal of gathering additional information on each TAA’s suitability for immunotherapy. In total, 117 [18,21,22,23,25,26,28,29,30,31,32,33,34,35,36,37,38,39,40,41,42,43,44,45,46,47,48,49,50,51,52,53,54,55,56,57,58,59,60,61,62,63,64,65,66,67,68,69,70,71,72,73,74,75,76,77,78,79,80,81,82,83,84,85,86,87,88,89,90,91,92,93,94,95,96,97,98,99,100,101,102,103,104,105,106,107,108,109,110,111,112,113,114,115,116,117,118,119,120,121,122,123,124,125,126,127,128,129,130,131,132,133,134,135,136,137,138] references from 1998 to 2020 were identified. Studies are composed of expression analyses by IHC or RNA sequencing (41), research into the TAAs’ role in oncogenesis (33), generation of specific T cells (31) clinical trials of T cell-based immunotherapy (6) and systematic reviews (6). Appendix A gives an overview of the studies identified for each TAA. Based on the information from the identified publications, as well as the expression data mentioned above, the TAAs were evaluated according to the weighted Cheever criteria and subcriteria [27], which were modified by exclusion of criteria targeting on clinical efficacy, in order to reduce bias towards clinically proven candidates. A total of ten points were available in the seven evaluation categories Immunogenicity (2.5), Oncogenicity (2.25), Specificity (2.25), Level of Expression (1), Expression on tumor Stem Cells (SCs) (0.8), Patients with TAA-positive tumors (0.6), and Number of Epitopes (0.6). Results are shown in Figure 2 and Appendix A. Most TAAs could be given a score in every category. There were 17 candidates where no information was found about their role in oncogenic processes, which lead to significantly lower scores, since 2.25 points could be obtained in this category. In 16 TAAs, the Expression on tumor Stem Cells could not be evaluated. Five TAAs were not allocated a score at all because the data in the literature were insufficient for evaluation, with four of them belonging to the group with medium expression in healthy tissue. Out of the 38 TAAs evaluated, eight candidates have reached a score of eight points and above (Figure 2). Seven of them were from the group of TAAs with low expression in healthy tissue and are cancer testis antigens.

CTAs CCNA1, for which an expression analysis in EOC was already performed by our group [25] and LY6K have shared the highest total score of 8.34 points, with maximum scores in all categories, except “Specificity” and “Level of Expression”. The other five top-rated CTAs CT45, IMP3, KIF20A, PRAME, and SP17 were similarly promising, and reached 8.07 points, due to a lower tier in “Expression on tumor Stem Cells”. Placed in third, MUC1 is the only TAA to reach over eight points in the medium-group and exclusively gets the maximum score in “Level of Expression”.

Medium-group antigen Survivin (7.91 points), alongside with low-group CTA MAGEA4 (7.7) complete the top-ten of highest rated TAAs, out of which only two antigens belong to the medium group. When compared to the low group TAAs, they collectively have a reduced “Specificity” Score because of their higher expression in healthy tissue. It is also noticeable that none of the candidates show absolute specificity to tumors, therefore the full points in this category were not attained by any of them. As we have already performed an in-depth analysis about CCNA1 expression and correlation to clinical features in EOC [25], in which we have identified CCNA1 to be a highly suitable TAA in EOC, we decided not to include CCNA1 in the following analyses. Because of promiscuous expression in healthy tissue, TAAs from the medium group pose an increased risk of on-target/off-tumor toxicity. As an additional safeguard for the identification of applicable targets, we have chosen only to include TAAs from the low expression group in IHC-analysis.

### 2.3. TAA Protein Expression in Clinical Specimen

To further analyze expression pattern of the TAAs on protein level and to validate the findings from the evaluation, the six highest ranked candidates out of the low group, namely LY6K, CT45, KIF20A, SP17, PRAME, and MAGEA4 were immunohistochemically stained in a clinically annotated tissue micro array of n = 58 EOC samples. Detailed results are provided in Appendix A. In the KIF20A TMA, five samples could not be assessed and were omitted from analysis. Staining intensity and homogeneity were independently evaluated by two pathologists. Representative IHC stainings are depicted in Figure 3a,b. There was a strong correlation between staining intensity and the percentage of positive cells for every TAA tested (*p* < 0.001). KIF20A emerged as an especially promising target, with positive staining in 53/53 samples, and 42/53 showing moderate (22/53) to strong (20/53) staining intensity. All samples presented a homogenous staining pattern with 100 percent conformity (Figure 3c). LY6K showed comparably strong results, with low to high staining in 54/58 samples. In total, 30/58 samples showed moderate staining intensity. One tumor was strongly stained. A total of 28 LY6K-positive samples showed a 100 percent staining, and only 5 were positive in fewer than 50 percent of the tumor cells (Figure 3c). IHC results for LY6K differed significantly from mRNA expression in the TGCA dataset, where over a third of samples were negative for LY6K. CT45 expression was observed in a third of the tumors (21/58), with nine samples showing high staining intensity. A total of 12/21 CT45-expressing tumors showed a 100 percent staining pattern, three samples were stained in under 50 percent of tumor (Figure 3c). To evaluate the combination of staining intensity (0–3 points from no staining to strong staining) and percentage of stained tissue, a Q-Score was calculated for each sample by multiplying the two factors (range 0–300). KIF20A, CT45, and LY6K achieved Q-Scores of 150 or above in 42, 14, and 20 samples, respectively. A Q-Score ≥ 150 shows that staining intensity for the tumor was moderate to high and at least 50% of cells displayed positive staining, making it a suitable surrogate parameter to differentiate between low-heterogenous and high-homogenous TAA expression. Expression results for MAGEA4 were similar to CT45. A total of 24/58 tumors were positive, with five EOCs presenting high intensity staining. However, only 11 tumor probes have shown positivity in under 50 percent of the cells, depicting a less homogenous staining pattern. A Q-Score ≥ 150 was reached in seven samples (Appendix A). The antigen PRAME was expressed in 38/58 EOCs but was showing low intensity staining in the majority of positive samples (26/38). Compared to the TAAs above, the staining pattern was less homogenous, with 24/38 tumors positive in fewer than 50 percent of the cells and only two samples with positivity in 80 percent or above. PRAME was the TAA with highest expression the TGCA dataset (Appendix A). For SP17, IHC-staining laid out a more restrictive expression with 5/58 positive tumors, with staining in fewer than 50 percent of the cells in 4/58 cases (Appendix A).

### 2.4. KIF20A Has a Wide Array of MHC-Presented Epitopes

We used the immune epitope database (IEDB) [139] to identify HLA-restricted epitopes of the seven TAAs that were identified to bind the MHC complex by mass spectrometry. An overview is provided in Appendix A. A total of 133 epitopes were identified for KIF20A, followed by PRAME, MAGEA4, and SP17 with 97, 36, and 12 epitopes, respectively. A total of 75 KIF20A epitopes have specifically been identified on cancer cells, with 4 epitopes described on EOC [140]. A total of 16 different KIF20A peptides were found to be presented in the context of HLA A*02:01, the MHC-class I protein with the highest prevalence of about 50% in the Caucasian population. The query for CT45 led to nine entries, with two epitopes being HLA-A2*01 restricted. For LY6K only one epitope was described, but that originating from an EOC sample [140].

### 2.5. KIF20A, CT45, and LY6K Are Prognostic Markers in EOC

In an exploratory analysis, expression of KIF20A, CT45, and LY6K as measured by IHC was correlated to clinical features to identify a potential prognostic or tumor biological relevance of the TAA in EOC. All patients had received cytoreductive surgery followed by carboplatin/paclitaxel in a palliative setting. Median age of patients was 60.5 years. Further patient characteristics are provided in Table 1. Clinical parameters assessed were overall survival (OS), progression free survival (PFS), time to progression (TTP) grade, age, and tumor stage at first diagnosis, presence of peritoneal carcinomatosis, lymph node or distant metastases, platinum sensitivity, and PDL1 expression. An overview of results is provided in Appendix A. For KIF20A there were no statistically significant differences in staining intensity or Q-Score in regard to the parameters (Appendix A). For tumors with CT45 Q-Scores of 0 and 1–149, we noticed a difference in the rate of platinum sensitivity, compared to tumors with Q-Scores ≥ 150 (Fisher’s Exact *p* = 0.039, Figure 3d). We also found a relation between higher T stage at first diagnosis with lower CT45 staining intensity (Fisher’s Exact *p* = 0.025, Appendix A) and lower Q-Score (Fisher’s Exact *p* = 0.004, Figure 3e) as well as the presence of peritoneal carcinomatosis (Fisher’s Exact *p* = 0.046, Appendix A). While there was no significant association between CT45 staining intensity and PFS (*p* = 0.16, Figure 4a), a CT45 Q-Score ≥ 150 was associated with prolonged PFS (*p* = 0.049; median 14.4 vs. 29.7 months, Figure 4b). We observed an association between lower LY6K Q-Scores and tumors in progressed T stages (Fisher’s exact *p* = 0.016, Figure 3e). Patients with tumors that showed moderate to high LY6K staining intensity also displayed an increased PFS (*p* = 0.041; median 12.6 vs. 27.5 months, Figure 4c). No association between KIF20A expression and PFS could be observed (Appendix A).

Given that the number of EOC patients analyzed by IHC was too low to produce reliable survival results, we additionally analyzed OS and progression-free survival (PFS) in dependent on KIF20A and LY6K expression in a dataset of Affymetrix HG-U133 microarrays deposited as online tool in KMplot [141]. CT45 was not represented by the dataset and could therefore not be analyzed. All calculations were based on EOC patients in stadium II-IV and dichotomized TAA expression data. High KIF20A expression was associated with reduced OS in all patients (n = 1074; *p* = 0.013) as well as only regarding patients with suboptimal debulking (n = 349; *p* = 0.034). PFS showed no significant difference. For LY6K expression, no significant difference could be observed in OS. However, high LY6K expression was correlated to prolonged PFS in EOC, after optimal debulking surgery (n = 177; *p* = 0.034). These data suggest that KIF20A and LY6K have an impact on tumor biology of EOC, reflected by their prognostic value independent of the clinical and tumor biological parameters correlated to the IHC data (Appendix A).

**Figure 3 ijms-24-02292-f003:**
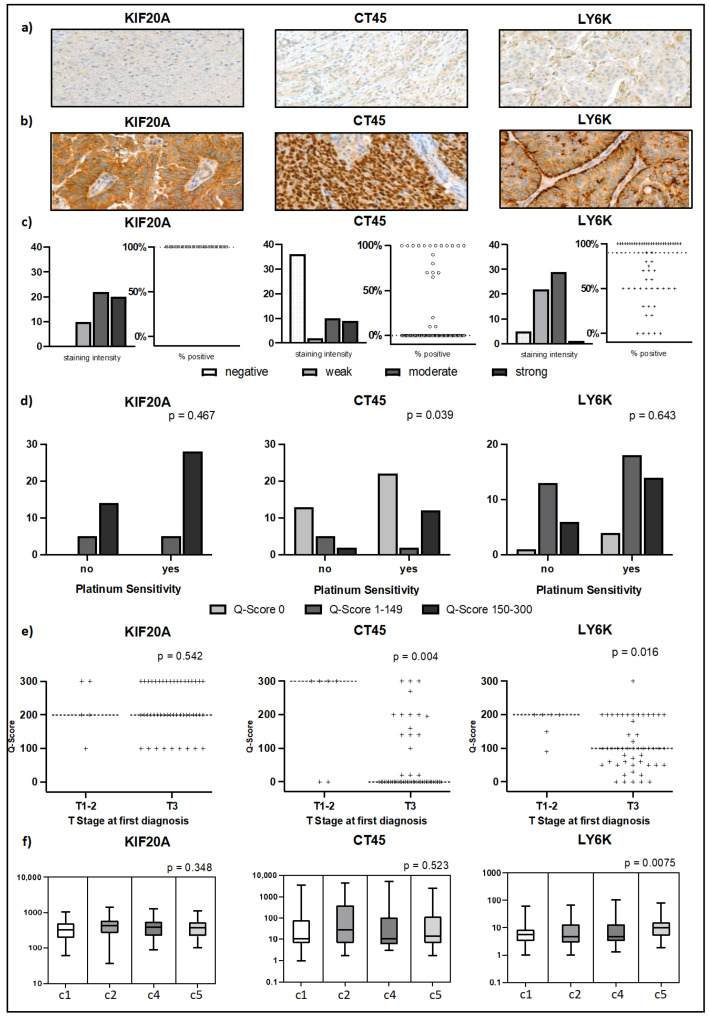
Results of IHC staining and clinicopathological analysis for KIF20A (left), CT45 (middle), and LY6K (right column): (**a**,**b**) representative stainings in tissue micro array of EOC, showing an example of weak (**a**) and strong (**b**) staining intensity; (**c**) results of IHC, showing number of samples with negative, weak, moderate or strong staining intensity (left) and percentage of tumor cells with positive staining for each sample (right); dotted lines mark median value (**d**) platinum sensitivity of KIF20A, CT45, and LY6K in relation to Q-Score of 0, 1–149 or 150–300; (**e**) Comparison of Q-Score values with T-stage of EOC at first diagnosis; dotted lines mark median value; (**f**) box plots depicting expression of KIF20A, CT45 and LY6K in different molecular subtypes c1 (high stromal response), c2 (high immune signature), c4 (low stromal response), and c5 (mesenchymal) of EOC, as defined by Tothill et al. [142]; boxes show interquartile range, median is marked by line inside the box, whiskers range from minimum to maximum value, *p*-values show results of Kruskal–Wallis test for differential expression.

**Figure 4 ijms-24-02292-f004:**
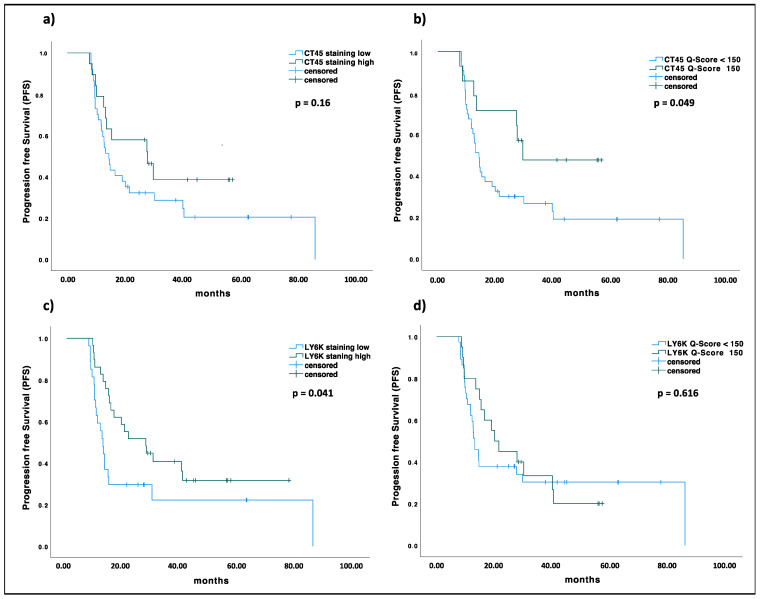
(**a**) Difference in Progression free survival in relation to staining intensity of CT45: blue CT45_low_ (negative to weak staining) vs. green CT45_high_ (moderate to high staining), median PFS CT45_low_ 14.39 months, CT45_high_ 27.76 months (*p* = 0.16); (**b**) Difference in PFS in relation to Q-Score of CT45: blue CT45 Q-Score < 150 vs. green CT45 Q-Score ≥ 150, median PFS CT45_Q < 150_ 14.39 months, CT45_Q≥150_ 29.7 months (*p* = 0.049); (**c**) Difference in PFS in relation to staining intensity of LY6K: blue LY6K_low_ (negative to weak staining) vs. green LY6K_high_ (moderate to high staining), median PFS LY6K_low_ 12.58 months, CT45_high_ 27.53 months (*p* = 0.041); (**d**) Difference in PFS in relation to Q-Score of LY6K: blue LY6K Q-Score < 150 vs. green LY6K Q-Score ≥ 150, median PFS LY6K_Q<150_ 13.18 months, LY6K_Q≥150_ 20.07 months (*p* = 0.616).

### 2.6. LY6K Is Differentially Expressed in EOCs with Mesenchymal Molecular Subtype

To further investigate a tumor biological relevance of the candidates in EOC, in an exploratory analysis we correlated KIF20A, CT45, and LY6K (Figure 3f) to the molecular ovarian cancer subtypes defined by Tothill et al. [142]. For this, the microarray data of the 285 tumors investigated in the original publication was assessed to extract RNA expression data of every TAA in the subtypes c1, c2, c4, and c5 (all cluster representing true malignant neoplasia). We tested statistically, whether the TAA were expressed differentially in the respective clusters. While no differential expression could be detected for KIF20A and CT45, we found that LY6K expression differed between the clusters (Kruskal-Wallis *p* = 0.0075, Figure 3f). Using multiple comparison, it was shown that LY6K expression was significantly higher in mesenchymal subtype c5, when compared to subtype c1 (Bonferroni-corrected *p* = 0.007) and subtype c2 (Bonferroni-corrected *p* = 0.017), but not to subtype c4 (Bonferroni-corrected *p* = 0.163).

## 3. Discussion

In this study, we provided a systematic identification and prioritization process for TAAs as targets for ACT in ovarian cancer. We identified several promising TAAs by in silico analysis and verified their suitability by IHC. With high scores during evaluation as well as high expression in a TMA of EOC samples, KIF20a, CT45, and LY6K emerged with the predescribed CCNA1 as the most suitable immunotherapeutic targets in EOC.

The identification of the TAA is the first and most crucial step in the development of any targeted T cell therapy. For selection of suitable TAAs in EOC, we focused on self-antigens. Using self-antigens as therapeutic targets poses the advantage of high inter-individual applicability, since their expression is observed in a high percentage of tumors. This allows the implementation of vetted and adapted therapeutic regimens that could be broadly used in a large number of patients. Neoantigens on the other hand, would allow the construction of a TCR that is specifically adapted to target the individual patients’ tumor and could thereby increase the anti-neoplastic effect of the therapy. However, this ‘personalized medicine’ approach has the limitation that each new intervention against a neoantigen has to be developed individually, which is not feasible in the majority of cases, given that neoatigen-specific TCRs have to developed and vetted not only with regard of functional avidity but also of potential TCR-specific toxicity.

We pursued a strategy of comprehensively collecting prediscribed TAAs and then analyzed expression databases as preselection mechanism to remove candidates that were deemed unsuitable for immunotherapy in EOC, either by high expression in healthy tissue, which leads to a increased risk for on-target/off-tumor toxicity or by low expression in EOC, as high expression of the TAA in target cells is one of the key predictors for clinical efficacy of TCR therapy. The preselection of TAAs also made a detailed literature search into the remaining candidates feasible. Upon this information, we then evaluated candidates regarding additional properties of “ideal TAAs” with predefined weighted criteria, thereby creating a reasonably objective ranking. The original evaluation criteria, defined by Cheever et al. [27] were established for the identification of TAAs as cancer vaccination targets. We modified the criteria to suit our approach of evaluaton for TCR-based treatment in EOC better. TCRs only recognize TAA epitopes that have been intracellularly processed are presented on MHC on the surface of the target cell. As this is the limiting factor for the TCR cascade to work, we decided not to evaluate the cellular location of expression of the unprocessed antigen. Further, we chose to omit the criterion of ‘therapeutic function’. As already discussed by the authors of the original prioritization study, there has been a bias towards clinically tested TAA candidates. In this project, we tried to prioritize candidates according to their expression features, their immunogenicity, and their oncogenic functionality in EOC to offer a guideline towards TAAs that are suitable for future immontherapeutic interventions. When more clinical data of in vivo immunogenicity and/or effectiveness of said interventions are available, we aim to reintroduce this criterion in the evaluation to filter out candidates that combine the cellular features of ideal TAAs with functionality as therapeutic targets.

Limitations of the selection approach are the restriction on predescribed TAAs, not allowing the identification of new therapeutic targets for tumor therapy, and the subjectivity in the evaluation process. Although criteria for evaluation were established in consensus by a panel of experts [27], they were applied to TAAs in EOC by different people, which could lead to varying scores, depending on the person performing the evaluation. By defining objective cut-offs e.g. for expression in healthy tissue, we tried to reduce inter-individual differences in evaluation. Furthermore, a TAA’s score still was afflicted by the literature provided. If a TAA was not investigated for its role in oncogenic and/or immunogenic processes, a high score would not be possible. This poses the risk of overseeing potentially suitable TAAs that have not been investigated as thoroughly and on the other hand could elicit selection bias towards candidates that are more established in immunotherapeutic research. Reviewing the results of the ranking, it was shown that cancer testis antigens (CTAs) form the group of the most encouraging candidates for an immunotherapeutic approach. CTA expression is generally silenced by promoter hypermethylation in normal tissue, with the exception of testis, which is considered an immunoprivileged site [143,144]. The expression pattern of a TAA, that is detectable in malignant but not in healthy tissue, reduces the probability of adverse events of ACT such as on-target/off-tumor toxicity. The risk reduction was the reason behind choosing only CTAs for the IHC staining after evaluation. Despite the higher scores, the TAAs MUC1 and Survivin have shown moderate expression in several healthy tissues, such as the stomach, lungs, or lymphocytes. We concluded that the slightly better performance in other evaluation criteria did not outweigh the risk of an immune response against non-tumorous tissue.

During the initial selection process, expression analysis of TAAs in healthy tissue and EOC was based on RNA sequencing data. Although mRNA and protein levels generally correspond, and the term ‘gene expression’ is often used in the context of measuring mRNA, there can be differences between mRNA and protein levels, caused by differences in post-translational or post-transcriptional regulation or protein degradation [145,146]. We included IHC data in cancer samples during the literature search and performed IHC ourselves on EOC probes, understanding protein expression as a key parameter. IHC staining showed negative or weak expression of three highly ranked TAAs (MAGEA4, PRAME, and SP17) resulting in their exclusion from further analyses. In this context, the question will have to be addressed whether the used antibodies against these candidates were the most suitable in terms of sensitivity. It can also be asked whether a thin slice of a tumor probe in the TMA sufficiently represents the tumors constitution, which is necessary to make qualified judgement on expression. RNA-based expression analysis may be able to better reflect a tumor’s heterogenicity. Further limitations of the IHC analysis may lie in the small sample size of the TMA. The remaining TAAs KIF20A, CT45, and LY6K emerged as promising candidates, alongside Cyclin A1, which our group has previously described as a suitable TAA in EOC [25,96,122]. Positive staining was found in all samples for KIF20A, 52/57 for LY6K, and over a third of probes for CT45. In our analysis, CT45 expression was lower compared to KIF20a and LY6K, with positivity in around one third of samples. This rate is consistent with IHC analyses from other authors [42,46,110]. KIF20A has shown high expression in EOC in the TGCA dataset on mRNA level, which was also the case in the IHC staining on protein level. For KIF20A, high expression is underlined by the large number of epitopes that have been identified to be processed and bound to MHC complexes in a wide array of different tumor cells. Interestingly, we observed differences between relatively low mRNA expression of LY6K in the ovarian cancer dataset and detectable LY6K in nearly all tumor samples investigated on protein level by IHC. This discrepancy can orginate from the fact that LY6K might have a long turn-over time, i.e. minimal transcription and translation is necessary to maintain a high concentration of the protein in the cells. However, degradation is correlated to both translation and presentation of respective epitopes, as reflected by low numbers of detected LY6K peptides in the ligandome analysis. This implies that protein expression is not necessarily the better marker to quatify visibility of the TAA to the TCR, while also indicating that the processing and presentation of a TAA could be observed as an independent quality criterion, which should be investigated more thoroughly in the future.

Expression of TAA in a high percentage of tumor cells ensures that a large amount of tumor mass can directly be affected by a specific TCR. This reduces the risk of cells escaping the immune response, creating a different tumor constitution, altering the tumor’s oncogenic capacities. The specific mechanisms of T cell-mediated tumor elimination are now studied in greater detail, and although a bystander effect of T cell killing of antigen negative tumor calls has been reported [147], the amount such effects in in vivo applications remains elusive. Therefore a high intensity and homogenous expression of TAA in tumor cells there is considered an important hallmark for the identification of suitable targets. In our IHC staining, KIF20A, LY6K, and CT45 have shown homogenous staining in a high number of tumors, thereby complying with the features mentioned above. The last limiting factor is the immunogenicity of potential targets. A noticeable discrepancy has been described between identified TAAs on the one hand and functional immunogenic candidates on the other [148]. Immunogenic potential of KIF20a has been proven by the creation of HLA-A2 restricted TCRs against pancreatic cancer [68]. For CT45, one study has also identified five HLA-class I specific epitopes of CT45 on EOC samples by immunoproteomics and has generated specific T cells for different peptides in a HLA-A3 and HLA-A11 specific manner [46]. Using such an approach, in which immunopeptidome of target cells is analyzed to identify peptides that are presented and processed by the tumor in vivo, could lead to highly effective and immunogenic TCRs.

The expression of TAAs can be an idependent prognostic factor of OS and PFS, and an association with unfavorable prognosis is a favorable feature for a TAA candidate, given that in case the TAA is relevant for maintenance of the malignant phenotype this impedes immunological selection of negative cell populations. We identified CT45 expression as a beneficial prognostic parameter for PFS as well as increased platinum sensitivity. The large-scale study from Coscia et al. reached similar findings. In our analysis of IHC staining and clinical data, we found no correlation between KIF20A expression and clinical or tumorbiological features. However, studies have shown that high KIF20A expression has been linked to reduced OS and high tumor grade, indicating a role in oncogenic processes [68,75,83,105]. We then used the Online Tool KMplot to correlate KIF20A expression to PFS and OS in a larger sample size, where KIF20A was significantly associated with reduced OS. As it was further shown that KIF20a knockout inhibits tumor proliferation in ovarian clear cell carcinoma [75], we concluded that KIF20A is a tumorbiological marker, associated with unfavorable prognosis. LY6K has shown a similarly high expression in EOC and we identified high LY6K expression as a prognostic marker for prolonged PFS. We also found an association between increased LY6K expression and the EOC molecular subtype C5, defined by Tothill et al. [142]. Mesenchymal subtype C5 has generally been characterized by lower expression of EOC differentiation markers and reduced OS [142], implying that LY6K represents a beneficial marker independent of molecular subtypes. As we have mentioned discrepancies between gene expression on mRNA and protein level, an analysis of molecular subtypes based on protein expression would be desirable for future research.

In conclusion, by using a systematic approach of evaluating and prioritizing TAAs, we have identified KIF20a, CT45, and LY6K to be highly suitable targets for targeted T cell therapy in ovarian cancer. To our knowledge, this research is the first systematic vetting of TAAs for EOC. Additionally, we are the first group to adapt the evaluation system of Cheever et al. and apply it to a specific tumor entity and modality of immunotherapy. KIF20A, CT45, and LY6K have reached high scores in the evaluation. We then validated these results by showing high expression of TAAs in a TMA of EOC samples. All three TAAs have been described as immunogenic, fulfilling the major hallmarks set for promising targets. We have shown that expression of the three TAAs was tied to clinical characteristics such as OS, PFS, stage, and platinum sensitivity. Especially KIF20a has stood out as a highly expressed protein in EOC with a vast number of epitopes that have been identified to bind the MHC. By identifying HLA-A2*01-specific peptides and generating specific T cells, immunogenicity of these TAAs should be further investigated for clinical application in the following steps. Furthermore, the systematic vetting approach can be utilized for the identification of TAAs in different tumor entities.

## 4. Materials and Methods

### 4.1. Identification of Candidate TAAs

A comprehensive list of predescribed TAAs was created by compiling entries of the databases “TANTIGEN 2.0” [149,150] and “Cancer Antigenic Peptide Database” [151], after exclusion of neoantigens. GTEx-Portal (accessed May 2020) was used to evaluate the candidates’ RNA expression in healthy tissue samples. For better visibility, TAAs were allocated in three groups with high, medium, and low expression. A low and medium level of expression was defined as <40 transcripts per million (TPM) and <400 TPM, respectively, in all analyzed tissue categories. Candidates with higher expression were excluded. In the next step, a TCGA dataset was used to examine TAA expression in n = 373 EOC samples via Human Protein Atlas (version 19.2, proteinatlas.org, accessed on 22 December 2022) [152,153]. Since TAAs in the medium group show higher expression in non-tumorous tissues and thus pose a higher risk for on-target/off-tumor toxicity than the restrictively expressed TAAs in the low group, different cutoffs were chosen. Using a detection threshold defined by the database, a TAA with a median FPKM value > 0.5 or average FPKM > 1 was as expressed in EOC [154,155,156]. TAAs with FPKM values under this threshold were omitted. Cutoffs in the medium group were median FPKM < 5 or average FPKM < 10, since expression above this threshold signifies robust expression [157]. A search of the MEDLINE Database via PubMed was conducted for each remaining antigen in May 2020. Search terms used were “x AND Ovarian Cancer” and “x AND Immunotherapy”, with x being substituted for each antigen. Studies with additional information on the suitability as cancer therapy targets such as expression analyses, previously conducted TCR-, CAR-, or vaccination trials as well as data on adverse events during trials were assessed.

### 4.2. Prioritization of Candidate TAAs

Prioritization of candidate TAAs was performed using weighted criteria and subcriteria as defined by Cheever et al. [27]. The criteria were modified, to better suit the evaluation of TAAs for TCR therapy in EOCs. TCRs can only recognize TAAs that have been intracellularly processed and presented in association with an HLA-molecule on the target cell’s surface, rendering the criterion ‘cellular location’ obsolete. To minimize bias due to differences in availability and/or mode of function of applied immunotherapy, categories based on clinical efficacy targeting the respective TAA were also excluded. The modified vetting criteria were normalized in a way that a maximum of ten points was attainable. The modified criteria and subcriteria with corresponding values are listed in Table 2.

### 4.3. HLA-Ligandome Data

Analysis of HLA-ligandome data was performed by searching the immune epitope database (IEDB) [139] for described epitopes of TAAs of interest. Only epitopes that were identified to bind the MHC-complex by mass spectrometry were included. Described epitopes were broken down by the tumor entities, where they were identified as well as association with HLA-subtypes.

### 4.4. Patients and Clinicopathological Features

A total of 58 female patients were selected from the ‘Tumor Bank Ovarian Cancer Network (TOC)’ tumor bank based on histology and initial treatment. All tumor specimens were collected before start of chemotherapy. All patients suffered from serous EOC and received cytoreductive surgery followed by platinum-based chemotherapy. Patients provided written informed consent for use of their biomaterial samples in biomarker studies. Consent was obtained using the standardized informed consent forms of the participating institutions. The project and consent process was approved by the ethic board of the Charité Hospital, Berlin (reference number EA2/005/14). All clinical and pathological features were extracted from the TOC data bank.

### 4.5. Analysis of TAA Expression by IHC

Top candidates were chosen for immunohistochemistry (IHC) in a tissue micro array of n = 58 EOC samples from the TOC tumor bank. Tumor specimens were cut in 4 μm sections and mounted on glass slides. After paraffin removal, hydration, heat-activated antigen retrieval in the DAKO-PTlink module (DAKO Glostrup, Denmark), and blocking of endogenous peroxidase activity by exposure to 3% hydrogen peroxide for 20 min, the slides were incubated at 4 °C overnight with corresponding antibody. Antibodies used were Anti-LY6K (ab224402), Anti-PRAME (ab219650), Anti-MAGEA4 (ab139297) (all Abcam, Cambridge, United Kingdom), Anti SPA17 (# PA5-58013, Invitrogen, Waltham, MA, USA), Anti-CT45 (HPA044757, Atlas Antibodies, Bromma, Sweden), and Anti-KIF20a (sc-374508, Santa Cruz Biotechnology, Dallas, TX, USA). Sections were processed with a Polymer HRP detection system (PV-9000, Zhongsam Company, Beijing, China). The slides were than stained with 3,3′-Diaminobenzidine and counterstained in hematoxylin. Healthy liver tissue was used as a normal tissue control for each antibody. Negative controls were carried out as above, omitting the primary antibodies. Staining intensity (0—negative, 1—weak, 2—moderate, 3—strong) and homogeneity (percentage of stained tumor cells) were evaluated at 400× magnification by two blinded pathologists independently. A Q-Score was calculated by multiplying the intensity with the percentage of stained tissue (Q-Score range from 0–300).

### 4.6. Molecular Subtype Analysis

For the exploratory analysis of potential association of the TAA candidates to a respective molecular subtype as described by Tothill et al. [142], the annotated Affymetrix HG-U133 Plus 2.0 microarray panel of that analysis was downloaded from the NCBI GEO database (accession GSE9899). Samples were normalized using invariant set method, expression data was exported as model-based expression equivalents (dChip 2.0 software) [1]. In case a TAA was represented by more than one probe set, the probe set with the highest average expression was chosen: 218755_at for KIF20A, 235700_at for CT45, and 223688_s_at for LY6K.

### 4.7. Statistics

The corresponding clinical data of samples in the tissue microarray was used to study relations of TAA expression with grade, stage, response to platinum, time to progression, and overall survival. Fisher’s exact test or Pearson’s chi-squared test were used to analyze contingency tables. Bivariate correlation was performed by calculating Spearman’s ρ. Survival analysis was calculated, using a log-rank test. The Kruskal–Wallis test was performed to evaluate differential expression in molecular subtypes. For post-hoc analysis, the Dunn–Bonferroni test was performed for multiple comparisons. All statistical analysis was performed in SPSS 28 (SPSS Inc., Chicago, IL, USA). All Figures were created using GraphPad PRISM (GraphPad Software, San Diego, CA, USA).

## Figures and Tables

**Figure 1 ijms-24-02292-f001:**
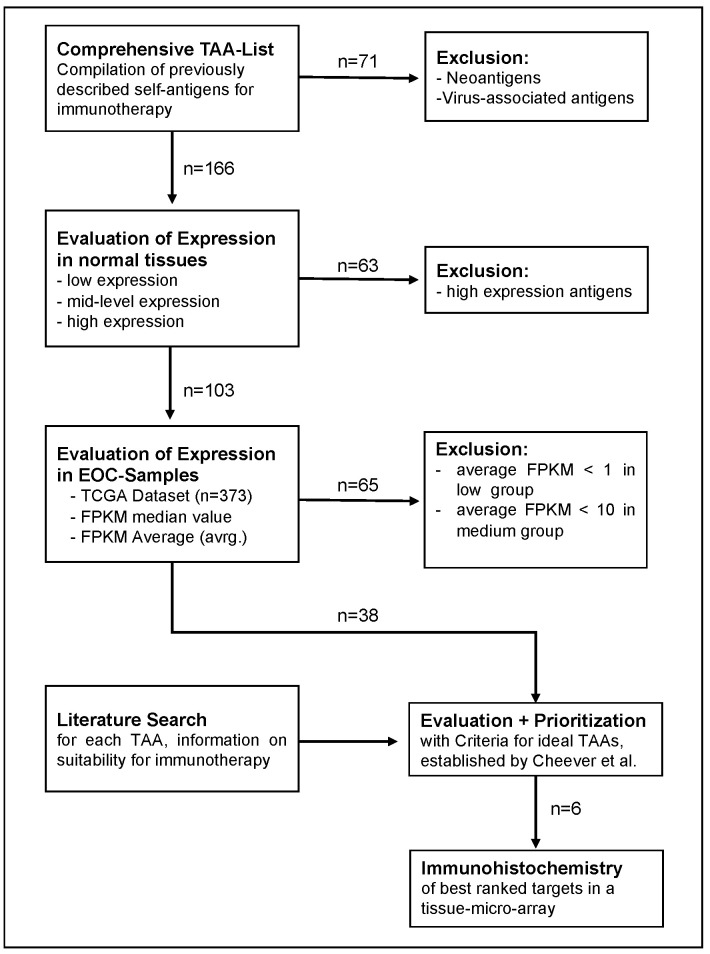
Flow Chart illustrating the process of TAA selection: after exclusion of TAAs with high expression in healthy tissue or low expression in EOC, 38 self-antigens were included in evaluation and prioritization, 6 targets were chosen for IHC.

**Figure 2 ijms-24-02292-f002:**
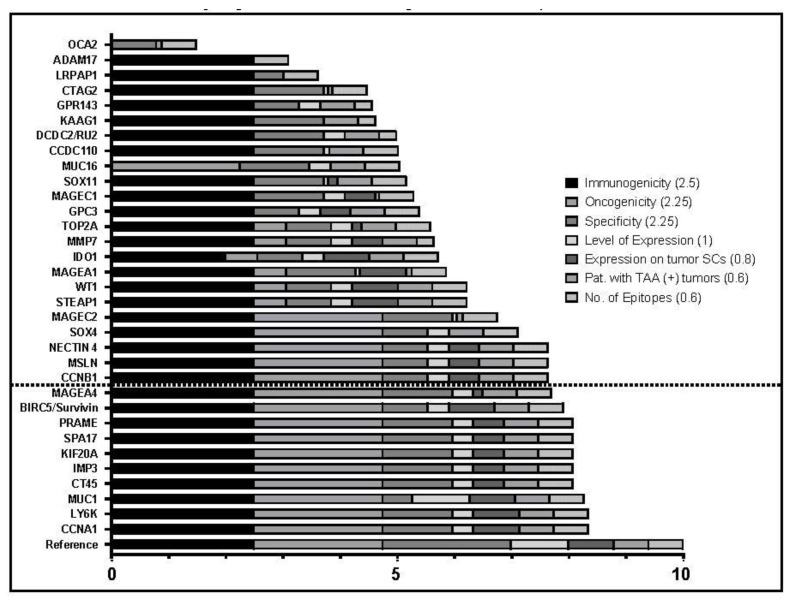
Bar chart depicting the results of TAA prioritization with weighted criteria for each TAA, modified after Cheever et al. [27]. A maximum score of 10 points was attainable. Legend shows color coded evaluation criteria and the maximum points attainable in brackets; Bottom bar shows a reference with the maximum points in each of the seven categories; Dotted line marks the ten highest rated targets.

**Table 1 ijms-24-02292-t001:** Clinico-pathological patient characteristics.

	*n*	*%*
Age ≤60 years >60 years	28 30	48.3 51.7
Histology serous	58	100
Grade		
2 3	6 52	10.3 89.7
Stage (FIGO)		
II III IV	3 46 9	5.2 79.3 15.5
Peritoneal Carcinomatosis		
no yes	7 51	12.1 87.9
Residual macroscopic tumor		
no yes	43 15	74.1 25.9
Primary Platinum response		
no yes unknown	20 37 1	34.5 63.8 1.7

**Table 2 ijms-24-02292-t002:** Evaluation criteria and subcriteria for the TAAs, modified after Cheever et al. [27] A total of ten points could be obtained in seven categories.

Criteria and Subcriteria	Definition	Pts.
Immunogenicity		2.5
HLA-restricted T Cell-Immunity verifiable	Experimental generation of HLA-restricted T cells, specific for a TAA is possible	2.5
T Cell-Immunity detectable in humans	Spontaneous T cell-Immunity against specific TAA is detectable in humans	2
Immunogenicity in animal models	Immunogenic in animal models observed, with similar antigen expression to humans	0.28
Antibodies detectable in humans	Antibody observed in humans (humoral response)	0.25
not applicable		0
Oncogenicity		2.25
oncogenic self-protein	TAA is associated with oncogenic process	2.25
persistent viral AG	persistently expressed viral antigen	0.77
Correlation with unfavorable outcome	Function of TAA unknown/uncertain, but expression correlates with unfavorable prognosis/decreased survival	0.56
tissue-differentiation, not oncogenic	TAA not oncogenic, but associated with tissue differentiation	0.27
stromal-Expression	Expression on tumor related stroma, but not on malignant cells	0.27
not applicable		0
Specificity		2.25
absolute Specificity	TAA is absolutely specific (e.g., mutated Oncogene, viral protein)	2.25
oncofetal AG	TAA is expressed in fetus with no or little expression in healthy adult tissues (e.g., cancer-testis-antigens)	1.22
overexpressed in Tumors	overexpressed in cancer, but expressed in some healthy tissues	0.79
abnormal posttranslational modifications	TAA expressed in normal tissues, but expressed in cancer with unique posttranslational changes (e.g., glycosy-, phosphorylation)	0.52
Tissue specific (expendable tissue)	Tissue specific expression in tissue relatively expendable for survival (e.g., prostate, ovaries)	0.47
Tumor stroma AG	normal TAA expressed on tumor-stroma	0.23
not applicable		0
Level of Expression		1
high, all cancer cells	Highly expressed on all cancer cells	1
high, most cancer cells	Highly expressed on most cancer cells	0.37
lower, all cancer cells	Lower level of expression on all cancer cells	0.23
lower, most cancer cells	Lower level of expression on most cancer cells	0.08
not applicable		0
Tumor Stem Cell Expression		0.8
Stem Cell Expression, presumptive	Evidence for expression on tumor stem cells	0.8
No info about SCs, but on all stages	Present at all stages of tumor development, from premalignant to metastatic lesions, but no info about stem cell expression	0.53
No info about SCs, but most cancer cells	Expression on most cancer cells, but no info about stem cell expression	0.16
not applicable		0
Patients with TAA-pos. Tumors		0.6
many Patients, high level	High level of expression in high fraction of patients in a tumor type	0.6
many Patients, lower level	Lower level of expression in high fraction of patients in a tumor type	0.1
fewer Patients, high level	High level of expression in lower fraction of patients in a tumor type	0.07
not applicable		0
No. of Epitopes		0.6
longer Antigen	Longer antigen with multiple (potential) immunogenic epitopes	0.6
short antigenic segment	Short antigenic segment with fewer (potential) immunogenic epitopes and potential to only bind to selected MHC-molecules	0.08

## Data Availability

Not applicable.

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
