# Peer review of "Target Selection for T-Cell Therapy in Epithelial Ovarian Cancer: Systematic Prioritization of Self-Antigens"

_ijms, 2023, doi:10.3390/ijms24032292_

Round 1

Reviewer 1 Report

The manuscript by Schossig et al. reported the development of systematic prioritization method for for ACT-based immunotherapies for epithelial ovarian cancer, and the identification of promising candidates, including KIF20A, CT45 and LY6K. This could provide valuable information for future experimental validation and clinical tests. The prioritization method may also be applicable to other cancer types. Below are my comments:

Page 5: “KIF20A emerged as an especially promising target, with positive staining in 53/53 samples, and 42/53 showing moderate (22/53) to strong (20/53) staining intensity.”

What are the thresholds for low/medium/high intensity?

Fig. 3c: meaning of the dotted lines in the right panel for each antigen? Also, for left panels I would suggest to color-code the expression levels in light-to-dark order.

“IHC results for LY6K differed significantly from mRNA expression in the TGCA dataset, where over a third of samples were negative for LY6K (data not shown).”

It is unusual to have positive staining results with negative mRNA results. Could it be due to difference in sample soruce or sensitivity of the two methods? Please discuss.

“CT45 expression was observed in a third of the tumors (21/58), with nine samples showing high intensity”

What is the definition of "high" here?

“KIF20A, CT45 and LY6K achieved Q-Scores of 150 or above in 42, 14 and 20 samples respectively.”

How was “150” determined and what is the significance of using this criteria?

“Although 9 samples were homogenously stained in 100 percent of the tumor, almost half of pos- itive samples (11/24) have shown positivity in under 50 percent of the cells.”

This sentence is confusing.

“Compared to the TAAs above, the staining pattern was less regular”

The defninition of “regular” is not clear.

Page 8 “the original TCGA microarray data was assessed to extract expression data of every TAA in the subtypes c1, c2, c4 and c5”

What is the reason for using TCGA data for correlation with subtypes study instead of using IHC data (protein level)? Authors have mentioned discrepancies between TCGA and IHC results.

“While no differential expression could be detected for KIF20A and CT45, high LY6K expression was associated with mesenchymal subtype c5 (Figure 3f).”

(for LY6K) Although the authors showed p=0.0075, the difference between c5 and other subtypes is quite minor from Fig. 3f. The meaning of p value shown here is not explained. Authors should show significant difference between c5 and other groups with appropriate correction method for multiple comparison.

Author Response

Dear Reviewer 1, 

We would like to sincerely thank you again for reading our manuscript and giving such constructive and helpful advice, which has helped us a lot. 

Please find attached a point-by point reply to all your comments. 

We will upload the revised version as soon as possible and we hope that the changes in the new version are to your liking.

Best wishes and have a nice weekend, 

Paul Schossig  

Reviewer 2 Report

While the paper is interesting and has value, there are certain elements that need to be improved.

The manuscript would benefit from some major restructuring - authors present results before the methods are discussed, which makes it hard to appreciate the results in the context of the paper.

Conclusions to the paper are lacking. Authors should include them as a separate section which will briefly address the novelty of the paper, the main achieved results and the limitations of the paper.

The novelty of the research is unclear in the current form, and it should be clarified and expanded on.

Additionally, some minor comments:
Figure 2 is of a very low quality - the text can barely be read.
The lines in Figure 4 are very hard to see.
Authors should avoid starting a subsection directly after a section title - a sentence or two describing the focus of the section could be useful.

Author Response

Dear Reviewer, 

Thank you for reading our manuscript and providing helpful and constructive criticism, which we have tried to implement in our revised version. 

Please find attached a point-by point reply to your comments. 

We have felt the same way points that you have raised, concerning the structure of the manuscript (e.g. Results before Methods) from the get-go. Unfortunately, we were not able to change the order of the manuscript, as we have to adhere to the guidelines set by IJMS. We even asked the editors to label our conclusion at the end of the discussion, but this was not possible either. 

Therefore, we tried to improve the text by adding some more methodical information in the Results section in order to make it more accessible and understandable for the reader. While we know this was not what you have in mind, we hope that you can see our efforts and think the issue to be at least partially resolved. 

I will upload the revised manuscript as soon as possible. 

Kind regards and have a nice week end, 

Paul Schossig 

Reviewer 3 Report

This manuscript is entitled: “Target selection for T-Cell therapy in epithelial ovarian cancer: Systematic prioritization of self-antigens,” the authors systematically investigated suitable tumor-associated antigens for targeted T-cell therapy of epithelial ovarian cancer. The authors evaluated the systematic vetting algorithm and identified KIF20A, CT45, and LY6K as promising candidates for immunotherapy in epithelial ovarian cancer. There are some questions and suggestions that may need to solve as below:

1.        More detailed content, such as weight, age, or gender, should be added to the data analysis of patients.

2.        Figure 4 does not show the difference in progression-free survival in relation to the staining intensity of KIF20A in patients with epithelial ovarian cancer. Please add to the manuscript and provide an explanation.

3.        The IHC data are vague, and the range is too small to judge the entire epithelial ovarian cancer area. The authors should adjust the data profile and mark the vernier bars on the data.

4.        The resolution of Figure 2 is insufficient, and the explanation for the reference below the author should be mentioned in the manuscript

5.        The data in the manuscript lack statistical significance evaluation and quantitative numerical analysis, such as platinum sensitivity of KIF20A, CT45, and LY6K that need to mark the groups relationship.

Author Response

Dear Reviewer, 

Thank you again for reading our manuscript and providing such helpful and constructive criticism, which has helped us a lot in improving our paper. 

Please find attached a point-by point reply to your comments. To the PDF I've attached a high-resolution picture of Figure 2 as well. We have also created a new supplemental file with additional information on the explorative data analysis, which I will upload as soon as possible. The same goes for the revised paper, which we hope is to your liking now. 

Kind regards and have a nice weekend, 

Paul Schossig

Round 2

Reviewer 1 Report

The reviewer would like to thank the authors for their efforts in addressing the comments. The reviewer believes the quality of manuscript has improved significantly and supports the publication in IJMS.

Reviewer 2 Report

I have reviewed the new version of the manuscript and the author reply. I apologize to the authors for providing the comment they were unable to follow due to the guidelines of the journal. Despite this, authors have still made the effort to make the changes and improve the flow of the manuscript. Beyond that, authors have addressed the minor comments I had.

Considering all of the points I have made initially were corrected by the authors, to the best of their ability, my recommendation is that the manuscript be published in its current form.

Reviewer 3 Report

The authors have resolved most of the concerns proposed by the reviewer, and the manuscript has been improved significantly. Therefore, we do not have further revision requirements for this updated manuscript.